# Optimization of the Surface Roughness Parameters of Ti–Al Intermetallic Based Composite Machined by Wire Electrical Discharge Machining

**Sonia Ezeddini** [1,2]**, Mohamed Boujelbene** [1,3,*]**, Emin Bayraktar** [1,*] and **Sahbi Ben Salem** [4]

1   School of Mechanical and Manufacturing Engineering, Supmeca Institute of Mechanics of Paris, 93400 Saint-Ouen, France; sonia.ezeddini@supmeca.fr
2   Development of Research, ALTRAN, 35136 Saint Jacques De La Lande, France
3   College Engineering of Hail, University of Hail, Hail 55476, Saudi Arabia
4   National Engineering School of Tunis, University of Tunis El Manar, Tunis 1002, Tunisia; sahbi_bensalem@yahoo.fr
*   Correspondence: mboujelbene@yahoo.fr (M.B.); bayraktar@supmeca.fr (E.B.); Tel.: +33-1-49452954 (E.B.)

**Abstract:** This work presents a comprehensive research using the Taguchi method and response surface methodology (RSM) to predict surface roughness parameters in wire electrical discharge machining (WEDM) manufacturing for a novel Ti–Al intermetallic based composite that was developed at Supmeca, a composite design laboratory for aeronautical applications in Paris, France. At the first stage, a detailed microstructure analysis was carried out on this composite. After that, the cutting parameters of the WEDM process were determined: Start-up voltage *U*, Pulse-on-time *Ton*, speed advance *S* and flushing pressure *p* were selected to find out their effects on surface roughness *Ra*. In the second stage, analyses of variance (ANOVA) were used as the statistical method to define the significance of the machining parameters. After that, an integrated method combining the Taguchi method and the response surface methodology (RSM) was used to develop a predictive model of the finish surface. The microstructure of the surface and subsurface of the cut edge, the micro-cracks, debris and craters and surface roughness of the specimens cut at the altered conditions were evaluated by scanning electron microscopy (SEM) and 3D-Surfscan.

**Keywords:** WEDM; TiAl-intermetallic based composite; surface roughness parameter *Ra*; response surface methodology RSM; taguchi method; ANOVA; *S/N* ratio

## 1. Introduction

Aeronautical engineering applications always impose new materials that meet their requirements, such as lightness, mechanical characteristics and machinability. Titanium Aluminide Intermetallic alloys are highly encouraging materials for meeting today's most prevalent demands for structural high-temperature materials. They are well known for their high strength and low-density characteristics and show high creep and oxidation resistance as well as high modulus and strength stabilities at high temperatures [1–3]. The potential applications of these alloys are in the aeronautic and aerospace industries.

TiAl-based intermetallic composites characterise an important class of high-temperature structural materials providing a unique set of physical and mechanical properties that can lead to substantial payoffs in aircraft engines, industrial gas turbines and in the automotive industry [4]. Due to these properties, Ti–Al based composites are principally used to produce the rotating parts of new turbines and the wheels of automobiles. Intending to understand the material's constituents, Bayraktar et al.

(2004) calculated the fatigue behaviour of ($\alpha_2 + \gamma$) Ti–Al intermetallics and revealed binary parts in structure composed of $\alpha_2 + \text{Ti}_3\text{Al}$ and $\gamma + \text{TiAl}$. In their work, a detailed fracture mechanism was interpreted with different plastic incompatibilities between the two phases [5]. However, the evolution of this intermetallic and its composition is still not fully understood.

This TiAl-based intermetallic is considered as a rigid material which has difficulties during the machining processes, for hard turning, etc. The aim for greater productivity, along with difficulties in the conventional machining processes, have led to the use of atypical cutting operations like electrical discharge machining (EDM). The EDM was utilized for cutting all types of conductive materials on account of its precision for the irregular and complex forms of the superficial EDM processes [6,7]. Ti–Al based intermetallic composites have high performance indices (Ip) and they are very attractive for aeronautic and aerospace engineering applications [8]. Our review of the existing literature has shown that there are no detailed works in the field of EDM machining.

The EDM machining of Ti–6A–4V was studied and optimized by different research groups [9–12]. However, there is no detailed research that treats the EDM machining of Ti–Al intermetallic based composite and its optimization.

Most recent reports have generally focused on the optimization of machining by the electrical discharge machining of steel and steel alloys, Inconel and Inconel alloys, aluminium and so forth. In these scientific works, the surface roughness parameters were optimized such as *Ra*. As a result, the desired surface finish was normally specified for only a certain application, and specific machining parameters were selected to achieve that desired level of finish. However, the EDM cut edge surfaces usually show different defects such as pores and micro cracks created due to the high-temperature gradient. Many factors may affect the surface finish in machining such as the cutting parameters, hardness of workpieces and the type of cutting tool. The machining parameters such as dielectric pressure flushing, pulse-on-time, pulse-off-time, duty cycle and cutting speed require careful selection to obtain the optimally tailored surface finish. In the age of global competitiveness, it is particularly important to optimize the machining parameters that can reduce machining cost and time and increase productivity to obtain the desired product quality. A variety of approaches have been used for optimizing and predicting surface roughness in electrical discharge machining [13–17]. The basis for those approaches is the theory of metal machining, experimental investigations, design of experiment (DOE) method and Taguchi and response surface methodology. The methodology for the experimental studies commonly follows models that relay the variation of the cutting parameters with surface roughness by means of regression analysis [18].

WEDM influences the quality of surface, the roughness of surface and making of the subsurface layer as reported for various materials [15]. A specific research work concentrated on the diverse performance measures in the EDM and the significant machining parameters [19]. In that work, the relationship control factors and responses were evaluated, like metal removal rate (*MRR*), surface finish (*SF*) and cutting width (*kerf*). The researchers used Taguchi's parameter design and significant machining parameters affecting the performance measures were identified as discharge current, pulse duration, pulse frequency, wire-speed, wire tension and dielectric flow. For these parameters, the genetic algorithm was employed to optimize the wire electrical discharge machining process with multiple objectives [15,18,19].

In another work, the multi-response optimization of cutting speed (*CS*) and surface roughness (*SR*) were evaluated using a utility method to find out the optimal process parameter setting [20]. The authors varied the experimental conditions of WEDM process parameters, such as pulse-on-time (*Ton*), pulse-off-time (*Toff*), peak current (*IP*), wire feed (*WF*), wire tension (*WT*) and servo voltage (*SV*) using pure titanium as the work material. It was confirmed that the most influencing factors on the surface roughness and the cutting speed were pulse-on-time, pulse-off-time and servo voltage dominating in cutting speed variation and pulse-on-time prevailing in surface roughness variation.

Besides this, response surface methodology (RSM) along with multiple linear regression analysis has been carried out to obtain the second-order response equation for the surface roughness parameter,

*Ra*, in EDM machining [21]. Pandey et al. established the possibility of accurately predicting arcing utilizing an ANN model of suitable architecture [22]. Another detailed analysis was conducted on the surface integrity, material removal rate and wire wear ratio for WEDM using grey relational analysis (GRA) and the Taguchi method [23,24].

The current paper studies in detail the effects of wire electro discharge machining factors on the cut edge surface of the titanium–aluminium (Ti–Al) intermetallic based composite that was designed and developed in our composite laboratory at Supmeca, in Paris, France. This composite was aimed to be tailored by cutting and machining as the final shaped specimen with an excellent surface quality in the French aeronautical industry. As part of a joint research project, an in-depth study was carried out on the surface roughness characteristics of the WEDM cut edge surfaces. For this reason, for an average surface roughness (*Ra*) value, for example, different WEDM operating factors such as Voltage (*U*), Pulse-on-time (*Ton*), Speed advance (*S*) and Flushing pressure (*p*) were varied along with the machining. Subsequently, an analytical model was developed using multiple linear regression and ANOVA analyses to predict the variation of the surface roughness as a function of these parameters.

## 2. Materials and Experimental Conditions

A detail microstructural analysis was executed on the Ti–Al intermetallic based composite. This composite is manufactured through an economical and efficient method, which is a combined method of powder metallurgy and thixoforming in the semi-solid state. Detailed information on the manufacturing process of this type of the composites was given in the former papers [25,26]. WEDM cutting tests were performed for defining the machinability of this composite. All the experimental tests were carried out on a WEDM machine and then the surface roughness values were measured employing 3D-SurfaScan in the laboratory. Detailed microstructural analysis were carried out by means of Optical Microscope (OM) and SEM for material characterization. The impact of cutting conditions on the surface of the Ti–Al intermetallic based composite was examined in detail. This cutting process was chosen for its frequent usage in aeronautical applications, and also because of the simplicity of its implementation and the absence of mechanical forces in the time of machining and the ease of observation and mensuration of the surface roughness on the cut edge.

### 2.1. Microstructural Assessment of the Intermetallic Composite

The design of the Ti–Al intermetallic based composite was carried out as part of a joint research project. It is known that titanium and its alloys have lower cutting behaviour compared to other light aeronautical materials; the cutting surfaces are easily altered or damaged during the machining operation. Difficulty in shaping is generally related to the metallurgical and mechanical characteristics of this element.

Nevertheless, the intermetallic alloy is known as one of the most difficult to machine. The critical nature of the shaped parts requires careful control of the integrity of the cut surfaces.

EDS-Chemical analysis and details of the optical microstructures together with a Ti–Al binary diagram are presented in Figures 1–3. In Figure 1, at the temperature of 700 °C, one can notice that the colour red depicts the phases of the intermetallic composite by using the binary diagram. The lamellar structure was observed with a lamellar spacing of 3 to 5 μm (Figure 2). Virtually no porosity was observed in the structure.

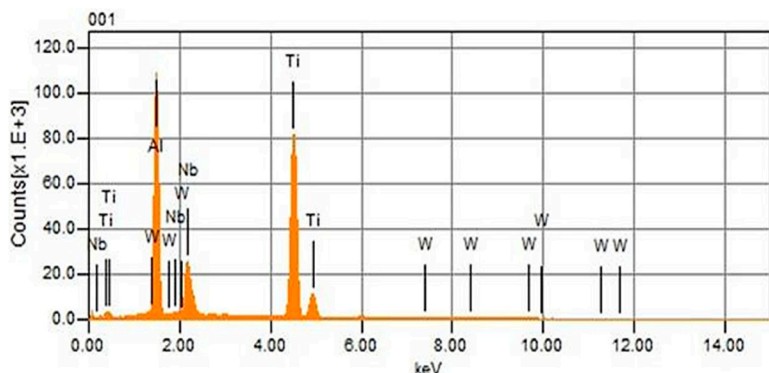

**Figure 1.** EDS, chemical analysis of Ti–Al intermetallic based composite.

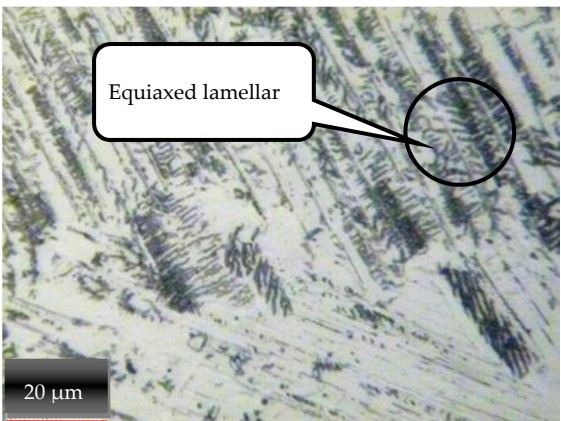

**Figure 2.** Microstructural evaluation of the titanium–aluminium (Ti–Al) intermetallic composite at the surface.

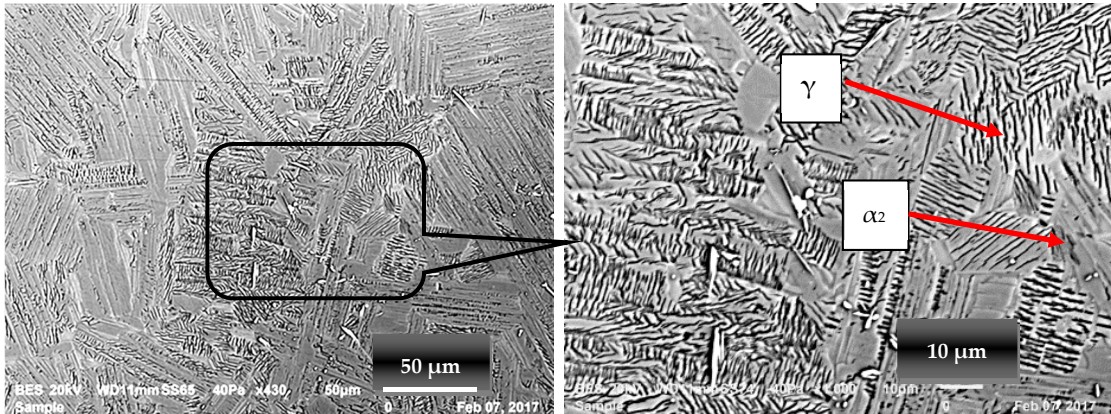

**Figure 3.** General microstructure of Ti–Al intermetallic composite at the thickness (transversal direction).

Through the analyses of the composition using the binary phase diagram, the typical microstructure of this composite can be considered as a mixed-phase "γ-TiAl and $\alpha_2$-Ti$_3$Al". Finally, the major microstructure is a substantially equiaxed lamellar structure made up of lamellar platelets containing platelets of the γ phase of the structure of the face-centred cubic lattice (fcc) (111) type *L10* and a very fine $\alpha_2$-(0001) hexagonal structure (Figure 3).

The constitution of this composite was executed as a part of a collaborative research project that is ongoing. The chemical composition of this studied composite is shown in Table 1.

**Table 1.** Chemical composition of Ti–Al intermetallic based composite.

| Composition | Ti | Al | Nb$_2$Al | Nb | Yttrium-Doped-Zirconia (Y-ZrO$_2$) | Mo | B | Zn–St |
|---|---|---|---|---|---|---|---|---|
| t% | 53 | 27 | 4 | 15 | 1 | 0.1 | 0.15 | 1 |

*2.2. WEDM Cutting Process and Cutting Parameters*

The WEDM cutting process is widely utilized in the aeronautic and aerospace, biomedical and automotive industries for the high-precision machining of all types of conductive materials such as metals, metal alloys, graphite or even certain ceramic materials of any hardness. The present research seeks to optimize the WEDM cutting conditions of the Ti–Al intermetallic based composite that is normally difficult to machinate with the conventional cutting processes.

The chemical composition of this composite is given in Table 1. For the cutting process, a plate with dimensions of 100 mm × 8 mm × 4 mm was cut to 9 samples. The experiments were realized on a wire rod erosion machine Robofil 190 (2002, Charmilles, Paris, France). The parameters of the process that were kept constant throughout the experimentation are:

- Workpiece: Ti–Al intermetallic based composite.
- Electrode (tool): 250 µm diameter of brass wire.
- Dielectric: deionized water.

We used the 3D-SurfaScan (2015-3D, STIL-BRANCH, Paris, France) to measure the roughness of surface *Ra*; the area examined was 2 mm × 1 mm. To study the linear effect of the process parameters and the level of the ranges of cut parameters, we investigated each process parameter at three levels. The values of the initial parameters were chosen according to the manufacturers' recommendations for the tested material. The selected process parameters, as well as their identified levels for single-pass cutting operation during WEDM of Ti–Al intermetallic based composite, is presented in Table 2. The proposed experimental design matrix planned as per "*L9*" orthogonal array (OA) for the current study is presented in Table 3. An integrated method was used to optimize the process which brings together the Taguchi method and the response surface methodology (RSM).

**Table 2.** Designation of the levels to the factors.

| Levels | Pulse-On-Time *Ton* (µs) | Servo Voltage *U* (V) | Speed Advance *S* (mm/min) | Flushing Pressure *p* (bar) |
|---|---|---|---|---|
| 1 | 0.8 | 80 | 29 | 6 |
| 2 | 0.9 | 100 | 36 | 8 |
| 3 | 1 | 120 | 43 | 10 |

**Table 3.** Machining parameters used in the experiment and results of cutting speed and breaking.

| Run | Control Factors and Levels | | | | Results |
|---|---|---|---|---|---|
| | *U* (V) | *Ton* (µs) | *S* (mm/min) | *p* (bar) | *Ra* (µm) |
| 1 | 80 | 0.8 | 29 | 6 | 2.46 |
| 2 | 80 | 0.9 | 36 | 8 | 1.92 |
| 3 | 80 | 1 | 43 | 10 | 1.8 |
| 4 | 100 | 0.8 | 36 | 10 | 1.7 |
| 5 | 100 | 0.9 | 43 | 6 | 2.53 |
| 6 | 100 | 1 | 29 | 8 | 2.01 |
| 7 | 120 | 0.8 | 43 | 8 | 1.74 |
| 8 | 120 | 0.9 | 29 | 10 | 1.74 |
| 9 | 120 | 1 | 36 | 6 | 2.96 |

A total of nine experimental had to be performed, using the combination of levels for each control factor as shown in Table 3. Four control factors including pulse-on-time (*Ton*), start-up voltage (*U*), feed rate or speed advance (*S*) and flushing pressure (*p*) for *Ra*, were selected, as shown in Table 2.

Therefore, the parameters of the process that were modified throughout the experiment are:

- Pulse-on-time (*Ton*): The time duration in which the spark (electron discharge) occurs between the electrode (wire) and the workpiece once the breakdown voltage of the dielectric is reached.
- Start-up voltage (*U*): The minimum input voltage value.
- Feed rate or speed advance (*S*): Feed rate of wire into the workpiece in mm/min.
- Flushing pressure (*p*): the pressure of injection of the dielectric.

The choice of the values of the levels of each parameter is related to the operating intervals of the WEDM machine.

## 3. Experimental of WEDM: Influence of Machining Parameters on the Surface Roughness

Arithmetic surface roughness (*Ra*) is the important constituent of the surface integrity (*SI*) of the cut pieces and the first qualification approach that should be considered. In WEDM machining, the machining factors that affect the surface integrity (*SI*) are generally the pulse-on-time, pulse-off-time, current, current tension, injection of fluid pressure and so on. Surface positions in this work should depend on the transformations produced during the thermal loading of the machining.

Particular attention should be paid to the choice of cutting conditions to ensure that the roughness of the cut surfaces meets the criteria given by the manufacturers. As well known, the WEDM cutting process is associated with high thermal stresses capable of inducing heterogeneous properties within the cutting surface of the workpieces. The alteration of these properties, generally at the surface or just under the surface of the manufactured parts, is referred to as surface roughness. The surface and the sub-surface are easily affected or damaged during the cutting operation. Cut surfaces thus have a strong influence on the performance of the components under the service conditions.

The feedback and analysis of the damage of dynamic components show that severe ruptures due to fatigue, creep and micro-cracking due to corrosion initiate and propagate systematically on or near the surface of the component, their origin being highly dependent on the quality of the cut surface [27]. Generally, the main problems due cutting process are identified as the burns, irregularities, edges or debris deposited on the surface, macro- and micro-cracks, cavities, micro defects such as inclusions, metallurgical alterations including distortion of the microstructure, phase transformations, temperature-affected layers and residual stresses.

### 3.1. Impact of Flushing Pressure and Tension on the Surface Quality

At the first observation from the experimental results, the injection pressure had a remarkable influence on the roughness of surface. The roughness values decreased from 2.8 to 1.75 μm, as shown in Figure 4, between two pressures from 6 to 10 bars; this is due to the phenomena of vibrations during the cutting process and/or the removal and erosion of the particles of material that were already removed.

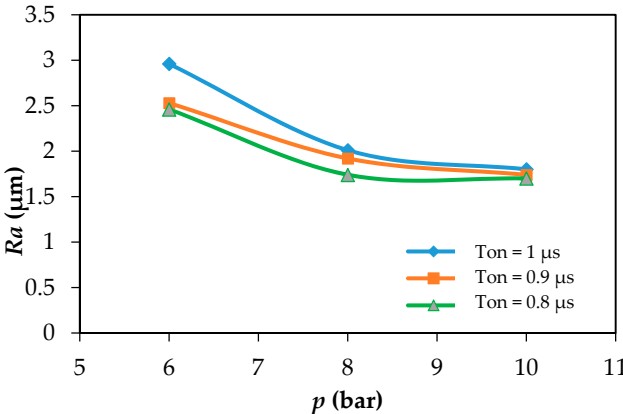

**Figure 4.** Curves of the effect of pressure injection on surface roughness *Ra*,

Injection pressure (flushing) usually helps to avoid any undesired contact between debris and the cut surface, which results in a better surface finish. However, in another study, the interactions of the pulse generator parameters were explained, such as on-time, off-time and servo-voltage and physical constraints such as wire tension, wire unspooling speed and injection pressure [28]. In the present work, the effect of these parameters was evaluated on the evolution of surface roughness in detail. As a result, the injection pressure of the fluid was found as another significant factor with a support level roughly at 33% [28].

Figure 5 presents the topographies of the 3D surfaces implemented on the Mountains software and Figure 6 shows the profiles of the 1D surfaces machined with certain cutting conditions. The surfaces cut with a lower pressure of $p = 6$ bar (Figure 5b) present great peaks and pits and comprise larger craters that are more numerous compared with the surfaces cut with a higher pressure of $p = 10$ bar which comprises less craters that are smaller in size, presenting the roughest profile.

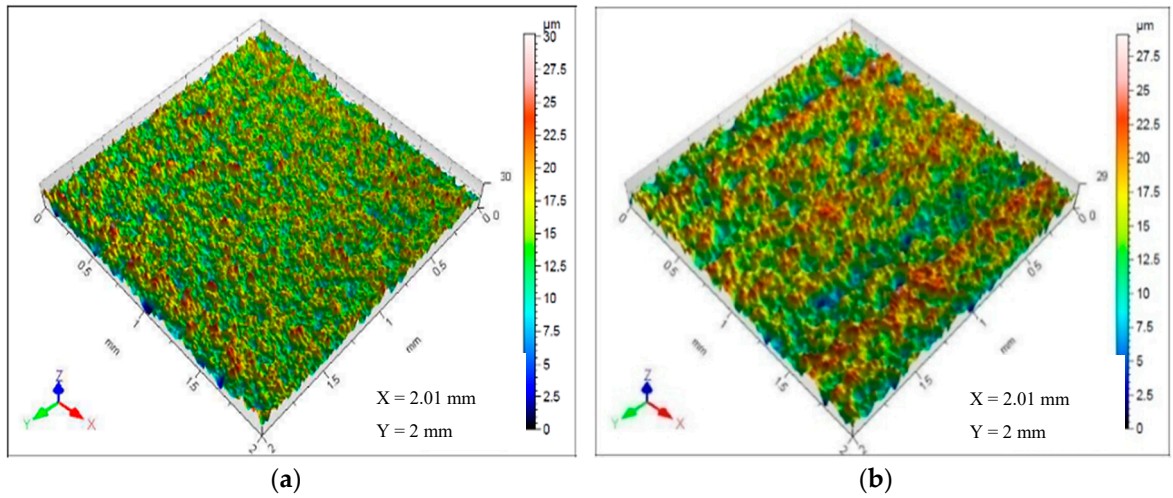

**Figure 5.** Effects of injection pressure on the surface topography: (**a**) $S = 36$ mm/min, $p = 10$ bar; $Sa = 2.32$ μm; (**b**) $S = 36$ mm/min, $p = 6$ bar, $Sa = 3.40$ μm.

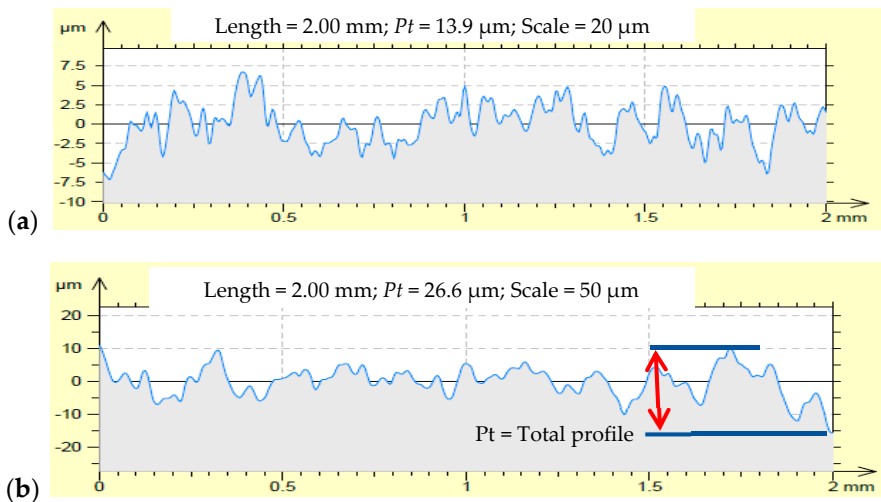

**Figure 6.** Effects of injection pressure on the profile of roughness: (**a**) $S$ = 36 mm/min, $p$ =10 bar; $Ra$ = 1.96 µm; (**b**) $S$ = 36 mm/min, $p$ = 6 bar, $Ra$ = 2.84 µm.

Research studies on flushing pressure reveal that it affects the surface roughness and tool wear rat, acts as a coolant and plays a vital role in flushing away the debris from the machining gap [29,30].

Figure 7 displays the surface quality of two WEDM-cut samples with different cutting conditions. The pictures of Figure 7 illustrate that the surfaces that are machined with the lowest pressure $p$ = 6 bar are the roughest compared with the surfaces cut with the high pressure of $p$ = 10 bar in a fixed value of the feed rate of $S$ = 36 mm/min.

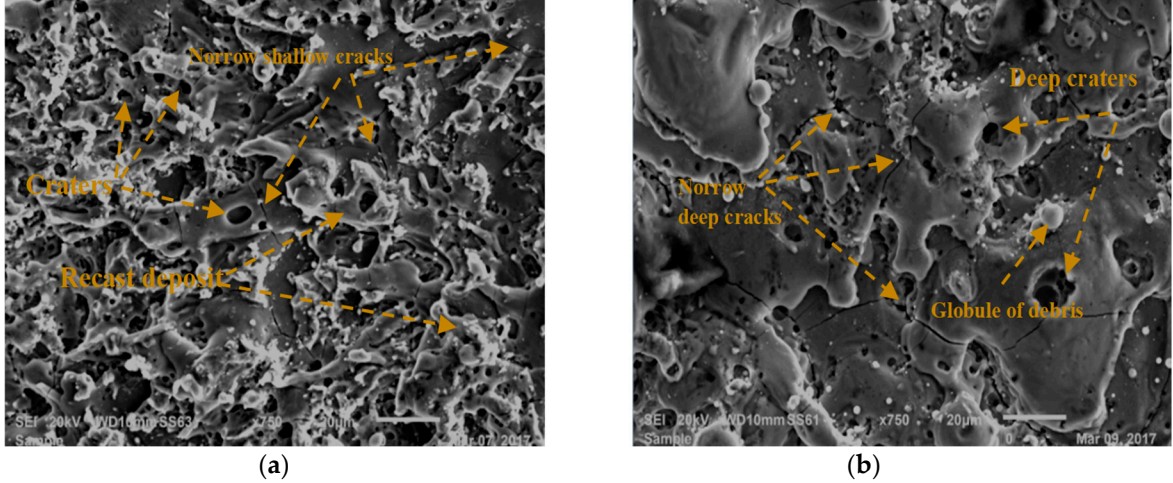

**Figure 7.** SEM observation of surface after wire electrical discharge machining (WEDM) as function of the cutting parameters; $S$ = 36 mm/min: (**a**) $p$ = 10 bars, (**b**) $p$ = 6 bars. Scale bar = 20 µm.

The surface has a contrasted appearance similar to the craters, resulting from the evaporation of the material during the machining, according to the SEM observation in Figure 7.

Figure 7a depicts a surface cut at a higher pressure of $p$ = 10 bar and cutting speed at the level of $S$ = 36 mm/min. It seems to have shallow cracks and very little debris. However, we can observe from Figure 7b the deeper craters and the larger irregularities at the surface of the specimen cut with a dielectric injection pressure of $p$ = 6 bar and cutting speed at $S$ = 36 mm/min.

The highest pressure helps to clean the surface and evacuate the fused material more efficiently during the cutting operation. However, the lowest pressure helps the energy to locate and spread to the bottom of the material, and subsequently causes the appearance of wider and deeper cracks with the presence of large craters.

Figure 8 presents the comparison between the surface roughness values cut with two cutting conditions $p$ = 6 bar and $p$ = 10 bar and gives the two roughness values of $Ra$ = 2.84 µm and $Ra$ = 1.56 µm.

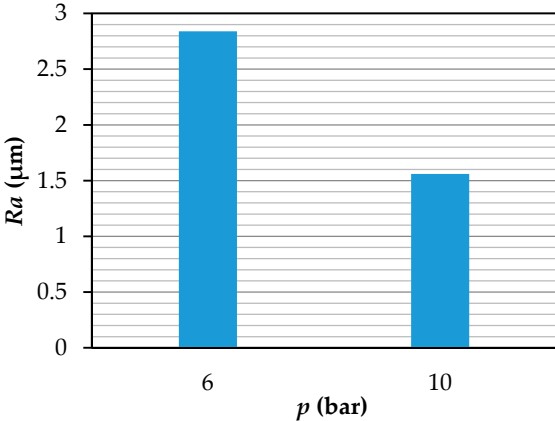

**Figure 8.** Value of surface roughness $Ra$ at $S$ = 36 mm/min, $p$ = 6 bar and $p$ = 10 bar.

Also, in case of the increment of the discharge energy and pulse-duration together, more melted material will be removed and deeper and larger discharge craters will be generated resulting in an increase of the surface roughness.

Figure 9 indicates the variation of the roughness values depending on the current voltage under the three different injection pressures. It seems that there is a significant influence of the voltage on the surface condition, especially with lower injection pressure. This relationship is related to the role of injection fluid that cools the cutting surface during the operation.

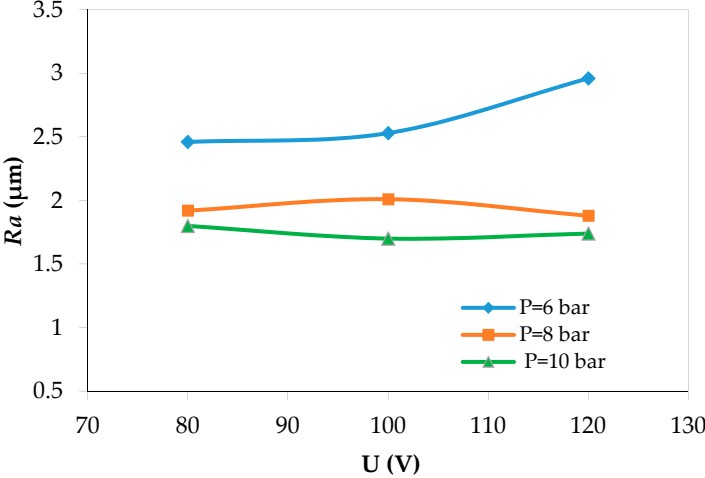

**Figure 9.** Evolution of the surface roughness $Ra$, depending on the current voltage.

### 3.2. Impact of Speed Advance and Pulse-On-Time on the Surface Quality

Figure 10 demonstrates the evolution of the surface roughness values, $Ra$, as a function of the cutting speed $S$. One may conclude that the $S$ value has an unremarkable influence on the surface roughness values, $Ra$. By increasing the cutting speed $S$, the roughness value remains mostly stable for the pressure values 8 and 10 bar, whereas for a lower pressure of $p$ = 6 bar, the surface roughness values decrease as a function of the cutting speed $S$ (29 to 43 mm/min).

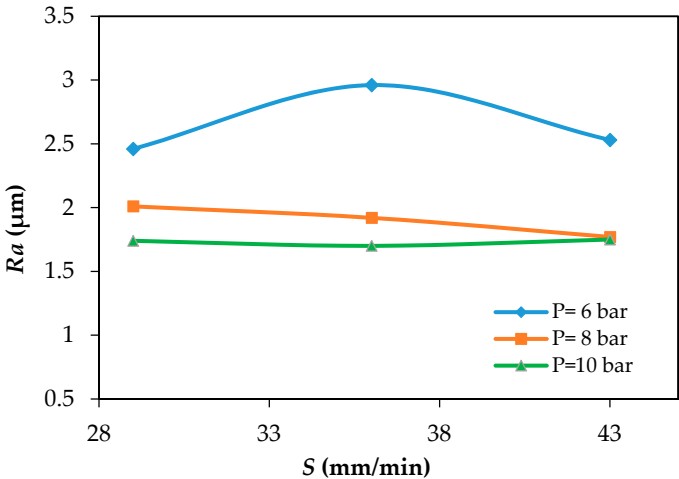

**Figure 10.** Curves of Effect of advanced Speed on surface roughness *Ra*.

These values are only experimental values carried out under the laboratory conditions and repeated 4 and 5 times. They are indicative values for the optimization of the cutting process.

Again, the microstructural analyses of cut surfaces of the samples carried out on SEM provides a very clear idea to explain the quality of the cutting of Ti–Al based composite intermetallics processed by WEDM.

Figure 11 displays the difference between a cut surface realized with a cutting speed equal to 29 mm/min and another cut surface realized with a cutting speed equal to 43 mm/min. The first one presents a smoother surface with a few amounts of debris, because the slowest cutting speed *S* allows evacuation time for adherent particles in the machined surface.

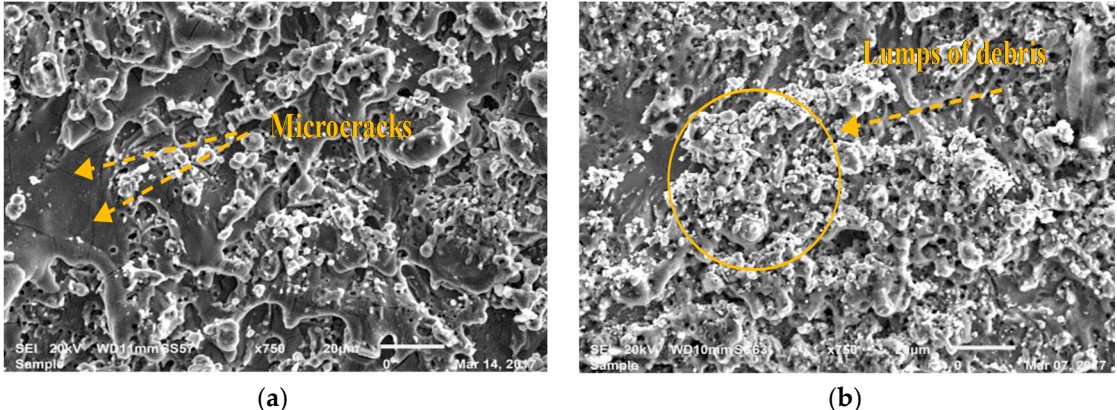

(**a**)　　　　　　　　　　　　　　　　　　　　　　　(**b**)

**Figure 11.** SEM observation of surface after wire electric discharge machining at cutting parameters; *p* = 6 bar; (**a**) Speed *S* = 29 mm/min; (**b**) Speed *S* = 43 mm/min. Scale bar = 20 µm.

Figure 12 depicts the cut surfaces processed with the lower value of the cutting speed of *S* = 29 mm/min and the higher value of the cutting speed of *S* = 43 mm/min by using a fixed value of injection pressure, *p* = 8 bar, to observe the influence of the cutting speed on the quality of the surface roughness values.

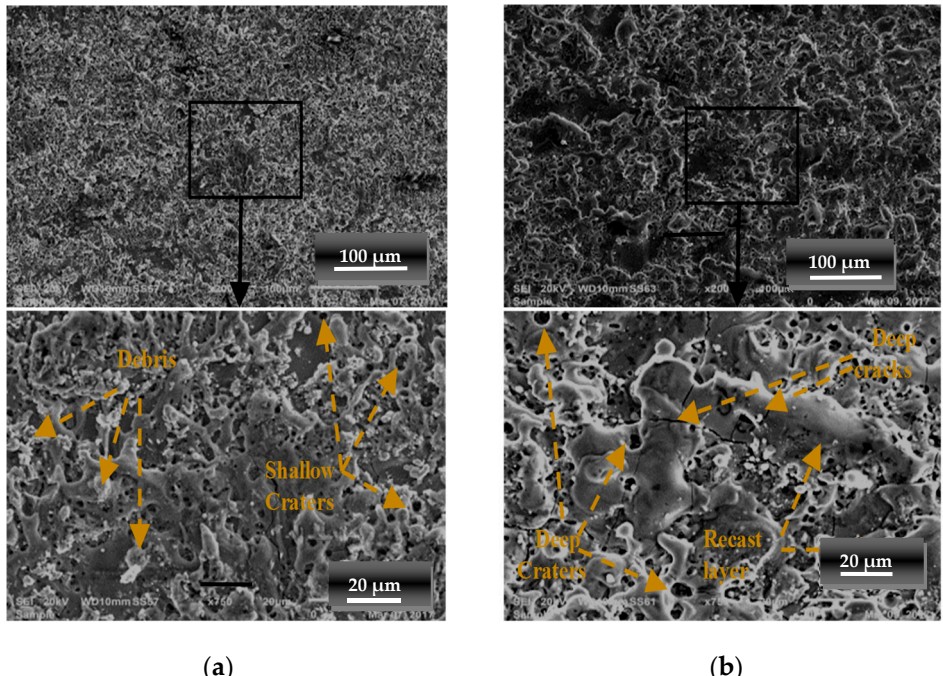

(a)          (b)

**Figure 12.** SEM observations of the cut surface after WEDM process at a pressure level of *p* = 8 bar under two different cutting speeds, (**a**) *S* = 29 mm/min and (**b**) *S* = 43 mm/min.

The cut surface obtained with a lower cutting speed of *S* = 29 mm/min comprises small craters and small debris compared to the cut surface with a higher cutting speed of *S* = 43 mm/min that comprises big craters, deep cracks and contain less of debris with rather smooth zones.

The cracking of the workpieces after wire electrical discharge machining (WEDM) was due to the energy applied of the wire, which affects the width and the shape of cracks. The cutting speed during the WEDM process is similar to a heat treatment called quenching which occurs during the cutting process. As such, the material is out of balance, which gives rise to micro-cracks at the cut surface. Two other factors that have an important influence on crack formation are the thermal conductivity of the workpiece and the hardness of the re-melted layer.

Figure 13 confirms the difference between the surface roughness value *Ra* with different cutting conditions; the surface roughness value is found to be around 2.01 μm in the case of *S* = 29 mm/min and *p* = 8 bar, whereas the roughness surface value is found to be 1.9 μm in the case of *S* = 43 mm/min and *p* = 8 bar. It seems that there is not a considerable difference between the two.

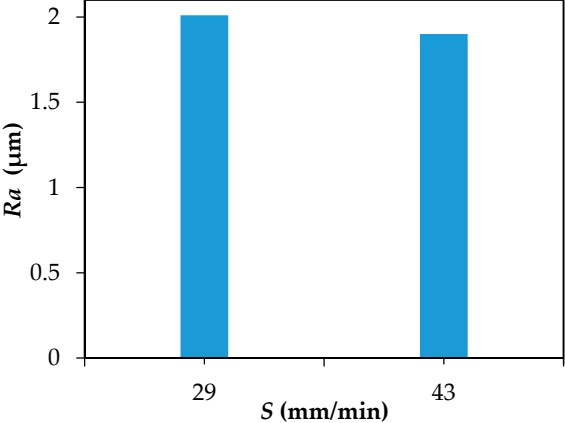

**Figure 13.** Diagram of the value of the surface roughness *Ra* at *p* = 8 bar; *S* = 29 mm/min and *S* = 43 mm/min.

On the other hand, pulse-on-time, *Ton* also affects the variation of the roughness and the state of the cut surface. If we increase *Ton*, the roughness increases in a significant way as observed in the Figure 14.

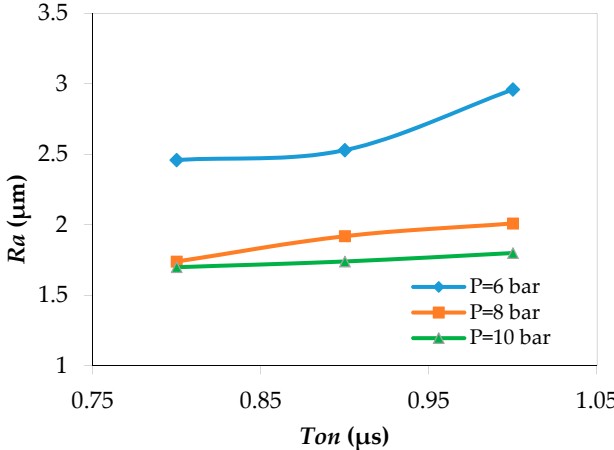

**Figure 14.** Effect of the pulse-on-time *Ton,* on the surface roughness *Ra*.

## 4. The Integrated Method

The present research project, which is ongoing, combines two methods to carry out an efficient analysis and to try to find satisfactory solutions, which are the Taguchi method and response surface methodology (RSM). In the first step, the Taguchi method was used to collect experimental data and to preliminarily analyse the link between the objectives and the parameters. Then, the RSM models were identified to predict the target functions so that a mono-objective model would be easy to obtain.

The standard experimental model based on Taguchi's orthogonal matrix "*L₉*" was used in this study and is presented in Table 3. This basic design uses up to four control factors, with three levels each of which are presented in Table 2.

Taguchi's design, developed by Dr. Genichi Taguchi, is a set of methodologies to account for the inherent variability of materials and manufacturing processes at the design stage. It is an alternative method to analyse losses that occur due to poor quality. It is also known as a robust design method to decrease noise sensitivity for any product or any process. The utilization of this methodology has become widespread in a of slew international industries. An advantage of the Taguchi method is that an investigation of numerous factors can be carried out simultaneously. Hence, not only can the controlled factors be examined, but also the noise factors. Despite its similarity to the design of experiment (DOE) method, the Taguchi method only stabilizes (orthogonal) experimental combinations, making the Taguchi design even more efficacious than a fractional factor design. By applying the Taguchi method, industries can adequately diminish the development cycle time of the product for design and production, and improve performance reliability and manufacturing efficiency, thus decreasing costs and augmenting profits. Additionally, the Taguchi design allows us to check the variability created by noise factors, which are generally ignored in the traditional DOE approach.

Many researchers have utilized the Taguchi method [28]. Farnaz et al. (2013) used a Taguchi *L18* plane to optimize wire tension, pulse time, current and tension to improve the integrity of surface, and carried out an optimization of material removal rate (MRR), and surface roughness value (*Ra*), using the *L25* [30]. Two primary tools in the Taguchi technique are the orthogonal network (OA) and the signal-to-noise ratio (*S/N*). The *S/N* ratio is adopted to determine the deviation of the quality characteristics from the desired values, including the highest Higher-The-Better (*HTB*), Nominal-The-Better (*NTB*) and Lower-The-Better (*LTB*). Nihat et al. (2004) considered the variation of *kerf* and the material removal rate by using ANOVA and *SN* ration [31].

In this investigation, we aimed to optimize the value of the surface roughness (*Ra*), so that the most objective type of function, "Lower-The-Better", (*LTB*) was utilized. The exact relationship between the *S/N* ratio and the signal is fixed by the equation below:

*LTB*: Equation (1)

$$\frac{S}{N} = -10 \times \log\left(\sum_{i=0}^{n} (1/Yi^2)/n\right) \tag{1}$$

where *n* is the number of experiments and $y_i$ is the value of the *Ra*.

The response surface method (RSM) is a set of mathematical and statistical techniques that provide adapted models between input parameters and responses to develop, improve and optimize a process [32–34]. The wide-ranging application of *RSM* is found in the industrial world, especially in the case where several input variables effect the performance measures or the product and/or process quality characteristics [33]. This work tried to find an appropriate approximation method for analysing the surface roughness relationship, *Ra*, concerning independent input parameters. A mathematical equation of the second-order polynomial response surface was used as illustrated in the following equation, where the coefficients of the function can be attained by the least-squares method:

$$Y = \beta_0 + \sum_{i}^{n} \beta_i X_i + \sum_{i<j}^{n} \sum \beta_{ij} X_i X_j + \sum \beta_{ij} X_i^2 + \varepsilon \tag{2}$$

Where *y* denotes *Ra*, *x* represents the parameters of WEDM ($T_{on}$: pulse time, *U*: start-up a voltage or servo voltage, *S* is feed rate or speed advance and *p* is the flushing pressure or dielectric injection pressure), $\beta_{ij}$ is the coefficients of each term and $\varepsilon$ is a residual error.

## 5. Analysis of the Results and General Discussion

As indicated in the previous section, the *S/N* ratio was utilized to identify the optimum parameters for the best surface roughness in WEDM cutting process of Ti–Al intermetallic based composite.

Table 4 presents the response table for means and signal-to-noise ratio for *Ra* of Ti–Al intermetallic based composite succeeding. This response table represents the impacts of diverse input factors on *Ra*. The higher slope in the main effects plots the corresponding values of the delta as higher in the response table. The rank precisely represents the level of the effect of the input based on the values of the delta.

**Table 4.** Response table for signal-to-noise ratios smaller is better and for means of *Ra*.

| Process Parameters | Level | Means | | | | S/N Ratio | | | |
|---|---|---|---|---|---|---|---|---|---|
| | | *U*(V) | *Ton*(μs) | *S*(mm/min) | *P* (bar) | *U* (V) | *Ton* (μs) | *S* (mm/min) | *P* (bar) |
| | L1 | 2.197 | 2.240 | 2.243 | 2.747 | −6.610 | −6.816 | −6.826 | −8.754 |
| | L2 | 2.150 | 2.070 | 2.243 | 1.927 | −6.599 | **−6.235** | −6.874 | −5.690 |
| Average value | L3 | 2.160 | 2.197 | 2.020 | 1.833 | **−6.489** | −6.647 | **−5.999** | **−5.254** |
| | Delta | 0.047 | 0.170 | 0.223 | 0.913 | 0.121 | 0.581 | 0.875 | 3.501 |
| | Rank | 4 | 3 | 2 | 1 | 3 | 2 | 4 | 1 |

Here, according to ranks, the effects of various input factors on *Ra* in the sequence of its effect are Pressure *p* and Cutting speed, sometimes called speed of advance, *S*.

This means that the pressure affects the *Ra* at the highest level and speed advance at the lowest level.

Figure 15 displays the effect of each cutting parameter on the ultimate surface roughness values: the pulse-on-time, the starting voltage, the cutting speed and the injection pressure. However, according to the curves of the effects, the pressure has the greatest influence on the roughness values, and the value $p_3 = 10$ bar corresponds to the smallest roughness value compared with $p_1$ and $p_2$. Additionally, the speed advance $S_3$ corresponds to the smallest roughness value regarding the values, $S_1$ and $S_2$, the $T_{on2}$ and $U_3$ corresponds the smallest roughness value compared with $T_{on1}$ and $T_{on3}$, $U_1$ and $U_2$ respectively.

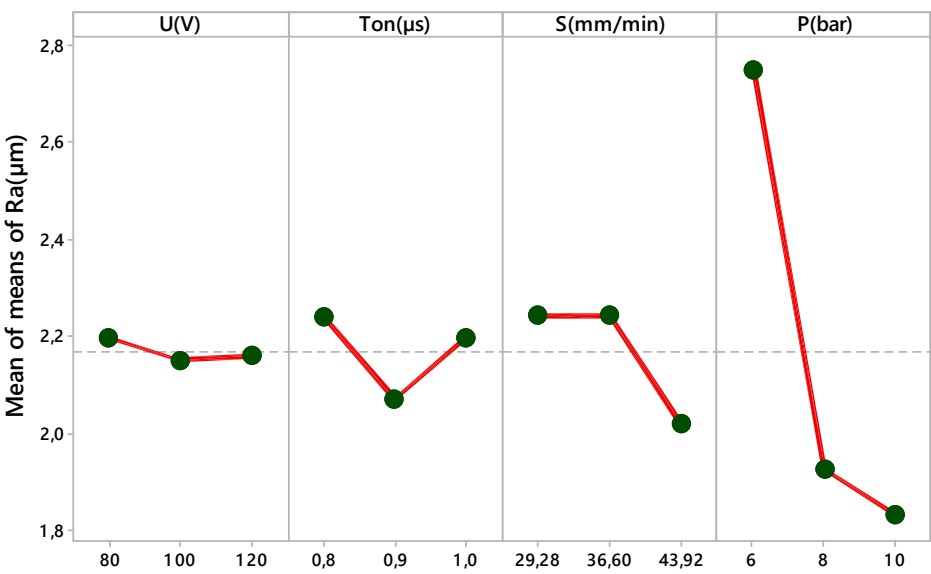

**Figure 15.** The influence of cutting parameters on surface roughness *Ra*.

At this point, the *S/N* ratio was used to determine the optimum parameters for a smaller value of *Ra* at the cut surface of the Ti–Al intermetallic based composite. From Figure 16 and Table 4, one can clearly observe that the optimal level of the cutting parameters is the level with the greatest *S/N* ratio.

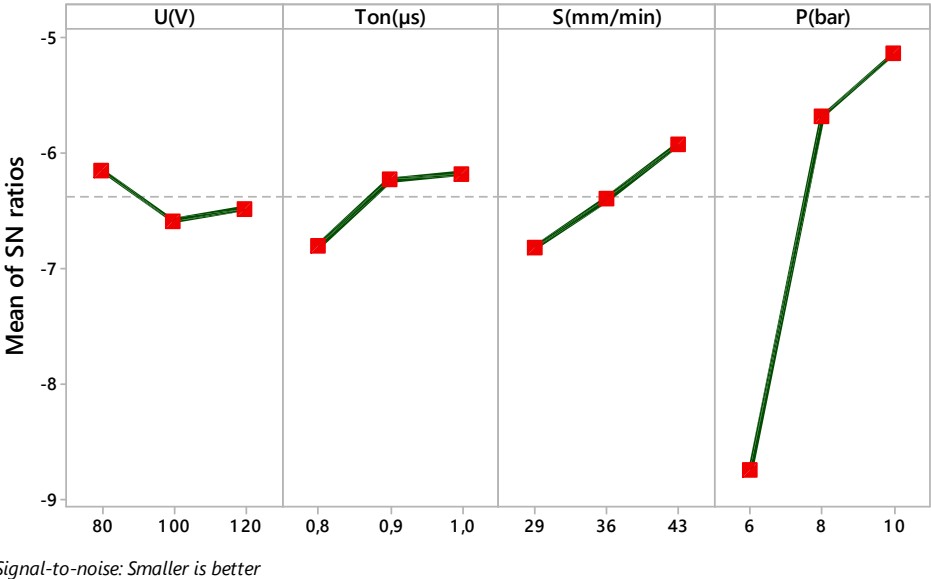

**Figure 16.** Main effects plot for *SN* ratios for *Ra*.

The comparative consequences of each factor have been evaluated through ANOVA of the surface roughness value.

The results of the ANOVA make it possible to see if the variables' explanations and their possible interactions give significant information to the model or not. The *F*-value, a ratio of the mean square of regression to the mean square error is used to prove the importance of each factor. All the *F*-values in the regression models exceed 0.05. The further significant coefficient is *R-sq*, which is defined as the ratio of the explained variation to the total variation and indicates the accuracy of the model.

Table 5 reveals that the pulse-on-time, flushing pressure and the speed of advance (cutting speed) are the most significant parameters, whereas the servo voltage is found to be less significant for minimizing the surface roughness. Further, it is also seen that the flushing pressure has a major

contribution in minimizing the surface roughness value followed by speed of advance during the cutting process of the Ti–Al intermetallic based composite.

**Table 5.** ANOVA result for *Ra*.

| Source | DF | Adj SS | Cont% | Adj MS | *F*-Value | *P*-Value | Remarks |
|---|---|---|---|---|---|---|---|
| Model | 8 | 2.32511 | 99.47 | 0.290639 | 148.62 | 0.000 | – |
| $U$ (V) | 1 | 0.01170 | 0.12 | 0.011697 | 5.98 | 0.037 | – |
| $Ton$ (µs) | 1 | 0.01881 | 0.17 | 0.018811 | 9.62 | 0.013 | Significant |
| $S$ (mm/min) | 1 | 0.13637 | 4.47 | 0.136368 | 69.73 | 0.000 | Significant |
| $p$ (bar) | 1 | 0.44436 | 74.73 | 0.444355 | 227.23 | 0.000 | Significant |
| $U^2$ (V) | 1 | 0.00934 | 0.10 | 0.009344 | 4.78 | 0.057 | – |
| $Ton^2$ (µs) | 1 | 0.02054 | 2,63 | 0.020544 | 10.51 | 0.000 | Significant |
| $S^2$ (mm/min) | 1 | 0.14188 | 1.49 | 0.141878 | 72.55 | 0.000 | Significant |
| $p^2$ (bar) | 1 | 0.32871 | 15.77 | 0.328711 | 168.09 | 0.010 | Significant |
| Error | 9 | 0.01760 | 0.62 | 0.001956 | – | – | – |
| Total | 17 | 2.34271 | 100.00 | – | – | – | – |
| Parameter | | | | | | | |
| $S = 0.0442217$; *R-sq* = 99.25%; *R-sq* (adj) = 98.58%; *R-sq* (pred) = 96.99% | | | | | | | |

The coefficient of determination *R-sq* (adj) reached 98.58% for a surface roughness *Ra* model. The *F*-value and *R-sq* indicate that the *RSM* model can be applied successfully as prediction models.

Figure 17 shows that the residuals are distributed approximately in a straight line, showing a good relationship between the experimental and predicted values for all the surface roughness *Ra* performances, and the variable follows the normal distribution. Therefore, the developed model is considered to be adapted to the observed values. Similarly, these figures show that the residues found are randomly dispersed but are independent.

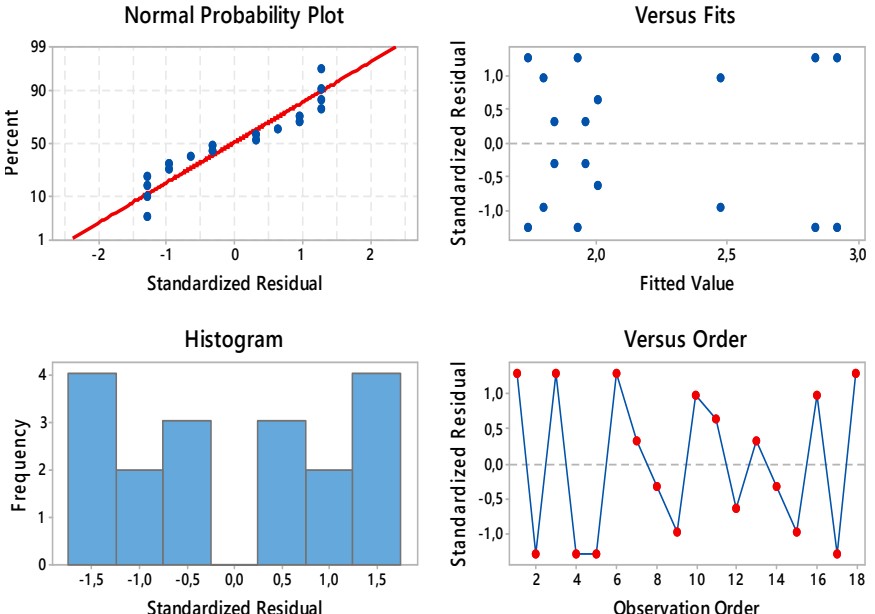

**Figure 17.** Residual plots for *Ra* value.

The second-order *RSM* model for minimizing the surface roughness value, *Ra*, is formulated under the following Equation (3). This equation represents the regression function for a surface roughness value *Ra* as a function of *p*, *S*, *U*, and *Ton*.

$$Ra = 7.15 + 0.0271U - 12.35\,T_{on} + 0.2717\,S - 1.3367\,p - 0.000121U^2 + 7.17T_{on}^2 - 0.003844\,S^2 + 0.07167\,p^2 \quad (3)$$

The *RSM* model is carried out with 9 experimental tests. For each combination of the input factors, the prediction value of the response, $Y_{j,\text{pred}}$, is compared with the experimental value of the response, $Y_{j,\text{exp}}$. Figure 18 illustrates this comparison between the experimental and predictive response values, relating the predictive and the experimental surface roughness values. As observed from Figure 18, the predictive results with *RSM* are very close to the experimental results.

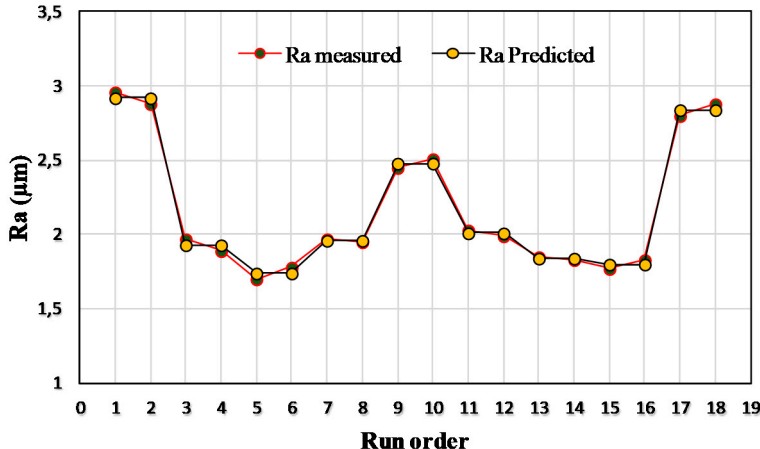

**Figure 18.** Comparison between measured and predicted values for Surface roughness *Ra*.

## 6. Conclusion

In the frame of the research project that is ongoing, RSM and Taguchi orthogonal array were utilized to identify optimum experimental parameter combinations. In order to find the minimum *Ra* values, it was important to specify the parameter combinations. Utilizing ANOVA analysis, the effective ratios of the parameters were determined.

Based on $R^2$ ratios, the validity of regression equations found with the RSM analysis for the prediction of roughness value was quite high ($R^2_{Ra}$ (adj) = 98.58%). From this consequence, these predicted functions can reliably be used for similar applications.

For recognizable security reasons, components of the aluminium-based composite used in aeronautics are sensitive parts and all precautions must be taken when shaping them, especially when cutting them by the wire electrical discharge machining (WEDM) process. A good surface roughness, *Ra*, is required.

Our concluding remarks are as follows:

- Several WEDM cutting tests were identified on the Ti–Al intermetallic composite on the orthogonal plane of Taguchi, "*L9*", by changing the cutting parameters (pulse time *Ton* (μs), voltage *U* (V), speed advance *S* (mm/min) and flushing pressure *p* (bar)). This means that three tests were considered for each factor, so many combinations were executed in this work.
- The optimum WEDM cutting conditions of (Ti–Al) titanium–aluminium intermetallic based composite is determined from the results of the signal-to-noise ratio in a Taguchi plane of different inputs on the surface roughness performance.
- The best surface roughness obtained were for the conditions below: a voltage of *U* = 120 V, a pulse time *Ton* = 0.9 μs, a wire feed speed *S* = 43 mm/min and a pressure *p* = 10 bar.

**Author Contributions:** Conceptualization, E.B. and M.B.; methodology, E.B.; software, S.E.; validation, E.B. and M.B.; formal analysis, S.E.; investigation, E.B.; resources, E.B.; data curation, S.E. and S.B.S.; writing—original draft preparation, S.E.; writing—review and editing, E.B.; visualization, M.B.; supervision, E.B.; project administration, E.B.; funding acquisition, E.B. All authors have read and agreed to the published version of the manuscript.

**Funding:** This research received no external funding.

**Acknowledgments:** The authors gratefully acknowledge the French aeronautical Society and our special thanks to G. Zambelis (research associate in Airbus-Paris) and H-A. Alhas (Improvement Manager, Airbus Defence, and Space-London) for their technical help and their discussion and valuable suggestions.

**Conflicts of Interest:** The authors declare no conflict of interest. The funders had no role in the design of the study; in the collection, analyses, or interpretation of data; in the writing of the manuscript, or in the decision to publish the results.

## Nomenclature

| | |
|---|---|
| EDM | Electrical Discharge Machining |
| WEDM | Wire electric discharge machining |
| *Ton* | Pulse on time (µs) |
| *p* | dielectric Pressure (bar) |
| *U* | Servo voltage (V) |
| *S* | Speed advance (mm/min) |
| *Ra* | Parameter of Surface Roughness (µm) |
| *Sa* | Parameter of3D Surface Roughness (µm) |
| RSM | Response Surface Methodology |
| ANOVA | Analysis of Variance |
| *S/N* ratio | Signal-to-Noise Ratio |
| *L9* | Taguchi design |
| *GRA* | Grey Relational Analysis |
| *ANN* | Artificial Neural Network |

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
