# Peer review of "Optimization of the Surface Roughness Parameters of Ti–Al Intermetallic Based Composite Machined by Wire Electrical Discharge Machining"

_coatings, doi:10.3390/coatings10090900_

Round 1

Reviewer 1 Report

  1. The article has a high scientific interest. Intermetallic materials and experimental methods are well explained. The cutting conditions and parameters used in the machine are correct. In the results, the 3D topographies and the images of the SEM microscope are of high quality, facilitating the interpretation of the results and validating the procedure used.
  2. The conclusions should be reviewed and completed. It has sufficient quality to be published.

Author Response

ANSWER:

Thank you for your kind evaluation and suggestion for our manuscript. We have revised and completed again according to your proposal.

Reviewer 2 Report

The paper reads well, but the following major edits should be covered prior to publication:

  1. Eliminating the overuse of "aeronautic" in the abstract. 
  2. Checking the English writing of the paper.
  3. Defining the process parameters (e.g., start up voltage, pulse on time, ......) in the methodology section. 
  4. Explaining why the authors choose the statistical levels mentioned in Table 2. 
  5. Illustrating the WEDM setup by showing a figure of the machine setup and workpiece. 
  6. Justifying why the authors choose an integrated method of Taguchi method and RSM, most probably from the literature.  Line 347, Page 12 "OA is used in the Taguchi method to save time and cost experiments", please elaborate. 
  7. In Figure 15, why surface roughness decreases then increases when increasing the pulse on time?
  8. Have the authors checked whether there is any material phase transformations after certain machining parameters?
  9. Why the authors usecd Ra only in the analysis, have the authors thought about involving other roughness parameters (e.g, RMS roughness).  For verification, have the authors tried the statistical model in other WEDM process conditions and/or different WEDM machine ?
  10. The main contribution and the novelty of this work should be presented in the conclusions section. 

Author Response

REVIEWER-II

The paper reads well, but the following major edits should be covered prior to publication:

ANSWER:

Thank you for your valuable evaluations and critics. We have revised and corrected according to your critics as follows:

Q1) Eliminating the overuse of "aeronautic" in the abstract. 

Ok, we have made it.

Q2) Checking the English writing of the paper.

Ok we have done it

Q3) Defining the process parameters (e.g., start up voltage, pulse on time, ......) in the methodology section. 

Ok, we have done in the nomenclature

Q4) Explaining why the authors choose the statistical levels mentioned in Table 2. 

The parameters chosen in table 2 are the optimal parameters for the proper functioning of the machine

Q5) Illustrating the WEDM setup by showing a figure of the machine setup and workpiece. 

We have done it

Q6) Justifying why the authors choose an integrated method of Taguchi method and RSM, most probably from the literature. 

The use of the Taguchi method is for the reduction of the number of tests and subsequently the reduction of cost. In addition, the taguchi method is the most usable by researchers in the field of mechanical manufacturing and even for the RSM method.

Q7) Line 347, Page 12 "OA is used in the Taguchi method to save time and cost experiments", please elaborate. 

We have done it

Q8) In Figure 15, why surface roughness decreases then increases when increasing the pulse on time?

Under laboratory conditions, all of the experimental results have been repeated at least 3 times and we have done as the mean values, we did interpret these results related to the experimental conditions.

Q9) Have the authors checked whether there is any material phase transformations after certain machining parameters?

In realty, certain disturbs on the cut edge in microstructure but we did not observe a considerable phase transformations.

Q10) Why the authors used Ra only in the analysis, have the authors thought about involving other roughness parameters (e.g, RMS roughness). For verification, have the authors tried the statistical model in other WEDM process conditions and/or different WEDM machine?

Actually, many research in this area, Ra is used for the results more confident. Many times, our industrial partners are mainly interested in Ra. It is true and interesting to make the statistical model for other WEDM conditions that we will consider planning to make it in our next work that is going on.

Q11) The main contribution and the novelty of this work should be presented in the conclusions section. 

Thank you, we have done it.

Reviewer 3 Report

Statistically, Taguchi's design and Response Surface Method (RSM) were used.
The progress of the experiment was well measured, but unfortunately evaluated and displayed.

  1. In equation (3) there are insignificant terms that act as noise and should have been removed.
    !!! Mainly Fig. 18 does not correspond to equation (3)!!! because all cuts can be at most parabola.
  2. Surely it cannot have two natural local maxima on one edge. It is necessary to remake the picture and specify how the parameters U (V) and Ton (μs) have been set, which are not displayed.
  3. What is novelty of this manuscript? Author should describe it in the end of Introduction. Author should describe used scanning electron microscope, optical microscope and SurfaScan.
  4. In the chapter "5. Analysis results and discussion" is really none discussion. Author should compare his results with other studies.
  5. Scales in all SEM figures are not readable. Fig. 5 should be in higher quality and text is not readable.
  6. All SEM figures are messy and descriptions in pictures are not readable. On SEM figures are visible cracks. Author should describe crack formation and visualize them in cross cut on metallographic samples.
  7. In Fig. 2 is no scale and i miss description, how was this metallographic preparate prepared and was etched?
  8. Fig. 1 (a) has to be in higher quality.
  9. Description in Fig. 11 is not english.
  10. Conclusion must be improved on the base of results in manuscript.
    Author should not use stars in equations. Author should evaluate cracks on the surfaces and say, if is machining possible without cracks formations.

Author Response

REVIEWER-III

Statistically, Taguchi's design and Response Surface Method (RSM) wereused. The progress of the experiment was well measured, but unfortunately evaluated and displayed.

ANSWER

Thank you for your kind evaluation and valuable critics. We have revised and corrected it according to your critics as follows;

Q1) In equation (3) there are insignificant terms that act as noise and should have been removed.

The equation 3 present good illustrations of the experimental phenomena of the Wire Electrical discharge machining of the Ti-Al intermetallic based composite.

Q2) Mainly Fig. 18 does not correspond to equation (3)!!! because all cuts can be at most parabola.

The text of the Fig. 18 has been changed, and we added the Pulse on Time Ton and Servo Voltage (U).

Q3) Surely it cannot have two natural local maxima on one edge. It is necessary to remake the picture and specify how the parameters U (V) and Ton (μs) have been set, which are not displayed.

The discharge voltage does not have a considerable effect on the surface quality of the parts obtained for the pressures P equal 8 and 10 bars, but for a pressure P equal at 6 bars, the roughness Ra increases significantly (Fig. 9).

The Pulse On time Ton has a small effect on the quality of the surface roughness of the parts obtained. When the Pulse On time Ton increases, the surface roughness Ra increases slightly (Fig 14).

Q4) What is novelty of this manuscript? Author should describe it in the end of Introduction. Author should describe used scanning electron microscope, optical microscope and SurfaScan.

The realization of the tests was using a WEDM machine, then we measured the roughness of each piece cut with a 3D surfascan, then we made a chemical attack on the surfaces that we wanted to see them using optical microscope and scanning electron microscope SEM which is a powerful and versatile tool for material characterization.

Q5) In the chapter "5. Analysis results and discussion" is really none discussion. Author should compare his results with other studies.

We have tried to reorganize this chapter we have improved our discussions

Q6) Scales in all SEM figures are not readable. Fig. 5 should be in higher quality and text is not readable.

The images in Figure 5 were obtained by SurfaSacn profile (for the measure the surface roughness), and we improved the quality of the images.

Q7) All SEM figures are messy and descriptions in pictures are not readable. On SEM figures are visible cracks. Author should describe crack formation and visualize them in cross cut on metallographic samples.

According to the researcher (Descoeudres 2006), the plasma develops between the 2 electrodes very quickly (50 ns) after the breakdown and stabilizes. The current remains relatively constant. The light emitted is particularly intense during the 500 ns then it weakens. Its intensity depends on the electric current.

For an air gap (Gap) of 10-100 μm, the discharge stimulates a volume whose diameter is of the order of 100-200 μm. This increases slightly during discharge. The current density is of the order of 40 - 100 A. cm-2. The voltage drop is located near each electrode on a layer 10 nm thick corresponding to the length of Debye with an intense electric field of the order of 80 - 100 V.cm-1. Steam bubbles are generated both in water and are due to the heat released by the plasma. 

The melting front of the electrode materials is decreasing. The two bubbles implode. Part of the metal still in the liquid state is ejected under the impact of the dielectric liquid which acquired a great kinetic energy during the implosion of the cavity.

EDM does not require direct contact with the workpiece, which eliminates the chance of mechanical stress as in the traditional turning, milling or grinding process.

Q8) In Fig. 2 is no scale and i miss description, how was this metallographic preparate prepared and was etched?

The scale of the Figure 2, has been presented

Q9) Fig. 1 (a) has to be in higher quality.

Figure 1a, has been canceled

Q10) Description in Fig. 11 is not english.

The description of the Figure 11 is corrected in English.

Q11) Conclusion must be improved on the base of results in manuscript.

We have done it again.

Q12) Author should not use stars in equations. Author should evaluate cracks on the surfaces and say, if is machining possible without cracks formations.

We have done it

Round 2

Reviewer 2 Report

  1. Accept, but check the English writing

Author Response

Dear Reviewer

Thank you for your kind attention,

As you suggested us, we have checked again English writing, the sentences and corrected typing errors.

Kind Regrads

E. BAYRAKTAR

Corrseponding authors

Reviewer 3 Report

I kindly ask Author to correct these comments:

- Fig. 1a is not readable and in poor quality
- Fig. 2 has no scale
- Fig. 3 scale i not readable
- Fig. 5 has not readable legend of axis
- Fig. 6 is deformed
- Fig. 7, 11 and 12 have no scale
-In equation (3) there are insignificant terms that act as noise and should have been removed. Mainly Fig. 18 does not correspond to equation (3)!!! because all cuts can be at most parabola.

Author Response

Dear Reviewer Thank you for your attention and valuable contribution for our manuscript

- Fig. 1a is not readable and in poor quality

We have now a readable  and qualty figure

- Fig. 2 has no scale

It has now

- Fig. 3 scale I not readable

Scale is readable now

- Fig. 5 has not a readable legend of an axis

It has a readable legend of an axis 

- Fig. 6 is deformed

Correcte dit

- Fig. 7, 11 and 12 have no scale

Thay have sacle now

-In equation (3) there are insignificant terms that act as noise and should have been removed.

Correcte dit

Mainly Fig. 18 does not correspond to equation (3)!!! because all cuts can be at most parabola.

Ok We have tried to correct this part

Kind Regards

E. BAYRAKTAR

Round 3

Reviewer 3 Report

In File

Author Response

Dear Reviewers,

Thank you again for your kind suggestions and critics of our research manuscript.

Additional of the corrections that we have tried to answer you in the last revision, this time, we have carefully corrected and improved certain points as you have proposed us.

Introduction, and experimental parts were revised in detail.

All the methods were described adequately.

All of the results were clearly presented and the conclusion were supported by the results. All of the important/basic corrections were put in yellow coloured in the text.

Finally English language were corrected by the language office of our school by the technical committee

Thank you for your attention

Kind Regards

Emin BAYRAKTAR

Round 4

Reviewer 3 Report

  1. Authors paid attention to the reviewers comments and improved the paper.

Author Response

Reply to Coatings Review of paper “coatings-681494”

Optimization of the surface roughness parameters of Ti-Al intermetallic based composite machined by Wire Electrical Discharge Machining WEDM

Responses

    1.This work cannot be accepted for publication until you write it in English.

Reply: Done

  1. Besides you should know that "p" stands for pressure, and >P< for power; moreover, say Table 3, what is >>-1<<, if it is to be MINUS one, so you should put a proper sign; once you write "1.7" and then >>1,74<<, etc.

Reply:Done

  1. Under 3.2, what do you mean >>On may conclude that “S” value have a...<<;

Reply:Figure 10 shows the evolution of the surface roughness values (Ra) as a function of the cutting speed (S). We conclude that the cutting speed value has a not so remarkable influence on the surface roughness values (Ra).

  1. page 13, you write >>These images can also explain...<<, it is rather ;

Reply: Done

  1. page 21, "U = 120 V", not >>U = 120V<<.

Reply: Done

  1. Concerning References, e.g. under 25. if you assume "P.G. Benardos,.." is right, so why do you put under 26. >>Enginsoy H, Gatamorta F,...<<, etc.

Reply: Done

  1. Benardos, P.G.; Vosniakos, G.-C., Predicting surface roughness in machining: a review. International Journal of Machine Tools and Manufacture 2003, 43, 833–844.
  2. Enginsoy, H.; Gatamorta, F.; Bayraktar, E.; Robert, M.; Miskioglu, I., Experimental and numerical study of Al-Nb2Al composites via associated procedure of powder metallurgy and thixoforming. Composites Part B: Engineering 2019, 162, 397-410.

    7. You should follow general rules of writing if it is to be reviewed and eventually considered for publication.

Reply: Done
